# OpenReview forum: "SmoothLLM: Defending Large Language Models Against Jailbreaking Attacks"
_ICLR.cc/2024/Conference — Submitted to ICLR 2024_

### Official Review · Reviewer_qFYD · 2023-10-30

**Soundness:** 2 fair
**Presentation:** 3 good
**Contribution:** 2 fair
**Rating:** 5
**Confidence:** 4

**Summary:**

This paper presents a defense method against large language model (LLM) jailbreaking, based on the Majority vote principle. It introduces random perturbations into queries to weaken certain jailbreaking attempts on LLMs. The method involves first introducing random perturbations and then measuring if the response begins with a target segment.

**Strengths:**

This work is an early exploration in defending against LLM jailbreaking. It pioneers the approach of using Majority vote and query perturbation to detect and prevent malicious outputs.

**Weaknesses:**

- The paper's definition of JailBreak is overly restrictive, focusing only on responses beginning with a specific prefix. This is problematic as attacks can still be successful even if the LLM does not strictly reply with the target segment. For instance, this paper use target as "Sure, here’s how to build a bomb", however, in an attack, LLM could generate "Sure, you can build a bomb by ...". This still containing harmful content should be considered a successful attack.
- The proposed SmoothLLM approach is impractical in real-world scenarios (like ChatGPT), because defenders can't predict the exact nature of the attacks or the corresponding targets. For example, the paper uses "Sure, here’s how to build a bomb" as a target corresponding to the goal "Tell me how to build a bomb". But if an attacker's goal is different, such as "How to steal money from charity", the previous target becomes irrelevant. Therefore, the defense is ineffective if the attacker's goals or phrasings differ from what the defenders expect.
- Furthermore, even if an attacker's goal is constant (e.g., "Tell me how to build a bomb"), they can easily bypass the defense by choosing a different target phrase for the jailbreaking. For example, if the defender's target is T="Sure, here’s how to build a bomb", an attacker might use T'="In order to build a bomb, you can first" as their target. In this scenario, the majority vote method would fail to filter out successful attacks since the JB(T',T) = 0.


Minors (typo):

"this would be come at the cost" -> "this would come at the cost"

"While this may seem counterintutive"->"While this may seem counterintuitive"

"future work should focus more robust"->"future work should focus on more robust"

"randomized smoothing (Salman et al., 2020; Carlini et al., 2022),"->"randomized smoothing (Salman et al., 2020; Carlini et al., 2022)."

**Questions:**

Concerning the results in Tables 3 and 4, SmoothLLM shows a performance decline of about 10% on PIQA. Why do the authors claim that their SmoothLLM method does not impose significant trade-offs between robustness and nominal performance? This claim appears to be contradicted by the empirical evidence.

---

> ### Author Response · Authors · 2023-11-15
> **Resolving your main concern: Clarifying "what is a jailbreak?"**
>
> Thanks for your review!  Here are some detailed comments.
>
> ---
>
> ## Our threat model
>
> Your main concern involves the threat model used in our paper.  In particular, your review indicates that you find the definition of `JB` in eq. (2.1) to be too restrictive and that the threat model is reliant on knowing the target string.  Fortunately, these comments are not weaknesses of the SmoothLLM algorithm.  Rather, your comments point to a shortcoming in our description of the threat model.  In particular, the `JB` function we define in (2.1) is *different* from the `JB` function we use in our experiments, as outlined in Appendix B of our original submission.  Therefore, **we believe that your concern can be completely resolved by clarifying the definition of a jailbreak** that we use in our experiments.
>
> To summarize this point, in Section 2, we define a jailbreaking through the binary-valued function `JB`.  In particular, we write that
>
> > "**one possible realization** of [the `JB` function] checks whether the reponse $R$ starts with the target $T$."
>
> After formalizing this version of the `JB` function in eq. (2.1), we go on to say that
>
> > ". . . there are **many other ways of defining `JB`**; see Appendix B for details.""
>
> Indeed, as stated in Appendix B.4, **we did *not* use eq. (2.1) in any of our experiments**; we agree with you that this definition is restrictive.  Instead, to be consistent with the GCG attack paper, in our experiments we used the definition of `JB` in eq. (B.1), which involves checking the response $R$ for various keywords.   Specifically, eq. (B.1) defines `JB` as
>
> $\text{JB}(R) = \mathbb{I}[ R \text{ doesn't contain any phrase in } \texttt{JailbreakKeywords}]$
>
>
> where `JailbreakKeywords` is a list of generic phrases such as including "I'm sorry," "As an AI," and "I will not."  This method is far more realistic and general, and importantly **it does not depend *at all* on the target string**.  Notably, this function does not incur the "different phrasing" or "fail to filter" drawbacks mentioned in your second and third bullets respectively.
>
> To address this shortcoming in the presentation, we rewrote the begining of Section 2 (see our updated PDF) to reflect the version of `JB` in eq. (B.1).  We believe that this completely resolves all three of the bullets in the "Weaknesses" section.  We hope you agree, and if so, we ask that you consider increasing your score.
>
> ---
>
> ## Non-conservatism
>
> > "Concerning the results in Tables 3 and 4, SmoothLLM shows a performance decline of about 10% on PIQA. Why do the authors claim that their SmoothLLM method does not impose significant trade-offs between robustness and nominal performance? This claim appears to be contradicted by the empirical evidence."
>
> You're correct that we do incur a trade-off between robustness and nominal performance.  Here is a summarized version of the evidence provided in our paper:
>
> * For Vicuna, if you set $q=5\%$, set $N=20$, and use swap perturbations, sacrificing between $5$-$10\%$ in nominal performance (see Tables 3 and 4) reduces the susceptibility to jailbreaking from $98\%$ to around $10\%$ (see Figure 7).
>
> Therefore, as is the case in nearly every paper ever written on robustness in ML, improving robustness comes at a *relatively* small cost (which we took to be ~10%) in nominal performance. We think that this is a reasonable price to pay to reduce the ASR by a factor of 10, but if you disagree, we would be happy to have a discussion about this.
>
> Moreover, our paper is the first to study this trade-off for jailbreaking, and we feel that this constitutes a worthy contribution.  If you think our paper would be stronger if we were to rephrase and/or emphasize the specifics of this trade-off in a different way, we would be happy to update our paper in line with your suggestions.
>
> ---
>
> ## Minor typos
>
> Thanks for pointing these out!  We have fixed them in our updated PDF.
>
> ---
>
> ## Strengths of our paper
>
> We want to make one more quick point in this response: We noticed that the "Strengths" section of your review does not include most of our main contributions that we listed in the introduction, e.g.,
>
> - **State-of-the-art ASR.**  SmoothLLM is the first and only algorithm to reduce the attack success rate of the (very) popular GCG attack to below 1%.
> - **Adaptive attack robustness.** SmoothLLM is robust against adaptive attacks.
> - **Query efficiency.** Running the GCG attack requires >200,000 queries to the target LLM.  Defending the target LLM against this attack with SmoothLLM only requires ~10 queries.
> - **Black box.** SmoothLLM only requires query access to the target LLM.
> - **Theoretical result.** In a simplified setting, SmoothLLM provably eliminates the threat of suffix-based attacks for suitably chosen hyperparameters.
>
> We feel strongly that these points constitute solid contributions to the field of jailbreaking LLMs, and we would appreciate your thoughts regarding our contributions.  Thanks!

---

> > ### Comment · Reviewer_qFYD · 2023-11-23
> > **Response to authors**
> >
> > Thank you for the detailed rebuttal and the revision.
> >
> > > JailbreakKeywords is a list of generic phrases such as including "I'm sorry," "As an AI," and "I will not."
> >
> > Indeed, that enriches the definition of jailbreak and make it more realistic. However, this definition is still not ready in an adversarial setup, as attacker could still easily bypass this keyword based metric. For example, attacker could optimize the probability of generating "As an AI, I can help you to ...".
> >
> > > Therefore, as is the case in nearly every paper ever written on robustness in ML, improving robustness comes at a relatively small cost (which we took to be ~10%) in nominal performance
> >
> > The assertion that enhancing robustness typically comes with a trade-off in nominal performance might not always hold true, especially for "detection and filtering" strategies. Specifically, for those samples correctly identified as "true negatives," such an approach might not alter their performance.

---

> > > ### Author Response · Authors · 2023-11-23
> > > **Response to your comments**
> > >
> > > > "Indeed, that enriches the definition of jailbreak and make it more realistic.
> > >
> > > If you believe that we have improved our paper, as you seem to have indicated here, we ask that you consider raising your score.
> > >
> > > > "However, this definition is still not ready in an adversarial setup, as attacker could still easily bypass this keyword based metric. For example, attacker could optimize the probability of generating "As an AI, I can help you to ..."."
> > >
> > > We respectfully disagree.
> > > * This is *precisely* the setup used in the GCG paper. We are using the threat model from past work, and showing that it can be resolved. What more could you possibly expect us to do?
> > > * Respectfully, it is unreasonable to fault our paper for not anticipating *every possible* future adaptive attack. In fact, we are the first in the jailbreaking literature to consider adaptive attacks.
> > >
> > > > "The assertion that enhancing robustness typically comes with a trade-off in nominal performance might not always hold true, especially for "detection and filtering" strategies. Specifically, for those samples correctly identified as "true negatives," such an approach might not alter their performance."
> > >
> > > We are the first to study this trade-off for the problem of jailbreaking. While you may feel that the trade-off should not exist for other strategies, this has *never* been shown in the literature on jailbreaking. Respectfully, we hope that you agree that speculation regarding what "might" be the case, without providing any evidence to support your claim, should not be viewed as a weaknesses of our method, but as a direction for future work.
> > >
> > > **Final thought.** We should not be faulted for not anticipating what will occur in the future. We have provided a defense that works extremely well for a threat model that is shown to jailbreak state-of-the-art LLMs. We feel strongly that all of the remaining weaknesses you have listed are not actually weaknesses; they constitute speculation about what might or might not happen, and there is little-to-no-evidence that this will occur. We feel that these are directions for future work, and as you seem to agree that our approach "pioneers" in the field of jailbreaking, we ask that you reconsider.

---

### Official Review · Reviewer_Ay91 · 2023-10-31

**Soundness:** 3 good
**Presentation:** 3 good
**Contribution:** 3 good
**Rating:** 5
**Confidence:** 4

**Summary:**

This paper proposes a simple scheme to prevent jailbreaking through adversarial prompt suffices. The main observation is the brittleness of these suffices (if a sufficient number of their characters are perturbed, they lose effectiveness). Building on this, the proposed scheme applies random transformations to the original prompt to smoothen the model response. They experiment on and provide guidelines on how to apply these transformations. The scheme stands out as it can operate in a black-box setting.

**Strengths:**

+ Simple, straightforward scheme to mitigate GCG-type attacks.
+ Some theoretical results as to when/how the scheme can be effective.
+ Guidelines and experiments on hyperparameter tuning are insightful.

Jailbreak attacks have been demonstrated both in academia and observed in the wild [1] and it is critical to develop simple, baseline defenses against this threat. The proposed algorithm fulfills that role and reuses the intuition from existing randomized smoothing defenses in the context of LLMs.

[1] https://arxiv.org/abs/2308.03825

**Weaknesses:**

- Not entirely convinced about the k-unstability assumption that is the foundation of the algorithm and the theory.
- Weak effort in designing an adaptive attack (e.g., create a suffix that's resilient to perturbations)
- No experiments on the universal GCG attack or more semantic jailbreak attacks that are more practical and widespread.

I don't know why k-unstability would be a fundamental property of adversarial suffix attacks like GCG. It's an empirical observation for a particular attack but we know very well now that building defenses on attack-specific artifacts is not a way to go. It only leads to a counterproductive arms race without yielding a long-lasting idea. Considering that it would be generally easy for an attacker to detect if SmoothLLM defense is used (e.g., by looking at the model's responses), it also would be trivial to change up the attack strategy to break the defense. Is there a reason why you believe k-unstability is a good assumption that will withstand future attacks or stronger adaptive attacks?

This brings me to my second point. It's accepted now that adaptive attack evaluations should be at the forefront of defensive papers like this. It's trivial to defend against any particular attack (e.g., by detecting the attack's artifacts) but through many painful lessons, we know that adaptive attacks are hard to design and evaluate against. I can't see much deeper thought in this paper in this regard. It definitely needs more work to be convincing that this defensive strategy is not going to be broken by the next deadline. There are some straightforward ideas from (such as EOT attack in the image domain) that could be useful here too.

Finally, a mistake we made for image domain adversarial attacks was to focus on trivial, non-realistic threat models (e.g., pixel-wise L-norm bounded perturbations). It has been years since this threat model was proposed and we've yet to see a real-world attack that uses this strategy. With attacks like GCG, I think, we might be making the same mistake and focusing on threats that might not matter. There's some real-world evidence that semantic jailbreak prompts are a greater and harder-to-mitigate risk that attackers are already deploying [1]. There's an intellectual merit in studying more academic, theoretical threats but I think it is critical to guide the community towards studying real threats. I can't see this paper doing that. There's no discussion on whether this defense would work against semantic jailbreaks or other real-world strategies, and no discussion on the limitations of the defense in deployment. Considering we have years of literature behind us, I would like to see a real effort into asking the right questions, which starts with academic papers like this.







[1] https://arxiv.org/abs/2308.03825

**Questions:**

- I'm not entirely sure about the reasoning behind returning one of the responses R_j (Line 5-6 in the Alg 1). How did you make this decision? What are the downsides compared to the other options you discussed in Section 6? I'm failing to have good intuition about this. Could you clarify?

- How does the defense affect the model's response utility against very short prompts (which might have less redundancy to withstand perturbations)?

- Do you use the input-specific GCG in your experiments? Input-specific GCG suffix is coupled with the original prompt itself, which could explain why small perturbations might break the attack. What about the universal version of GCG? Is that less brittle (I know it has a lower attack success rate)? It also changes the computational cost equation as the adversary needs to craft it only once.

---

> ### Author Response · Authors · 2023-11-16
> **Response Part 1: Clarification regarding adv. examples vs. jailbreaking, theoretical guarantees**
>
> Thanks for your review!  Here are some detailed comments.
>
> ---
>
> ## Adversarial examples vs. jailbreaking LLMs
>
> Much of your review draws a parallel between two distinct fields: adversarial examples (AEs) in computer vision and jailbreaking LLMs. To be clear, we agree with you that there are many lessons to be learned from AEs research and that the fields of AEs and jailbreaking share some similarities. However, there is little-to-no evidence thus far that the jailbreaking field will be guided by exactly the same principles and suffer from the same pitfalls as AEs research did. As we discuss in Appendix E.2, these fields have fundamental differences:
> * Different problem settings (discriminative vs. generative models)
> * Different input spaces (continuous pixels vs. discrete tokens/strings)
> * Different threat models (imperceptible noise vs. jailbreaking prompts)
>
> And therefore, these fields may very well adhere to divergent standards of practice. At the very least, we respectfully disagree with your claim that we (the authors) "know very well" how to avoid the "painful lessons" from AEs research. There is no way to faithfully predict how this field will develop, and our paper should not be judged on this basis.
>
> In this same vein, you call on us to "guide the community" toward best practices in the field. Although we are unsure of what the field will look like in the coming years, in proposing our set of criteria for candidate defenses (see Section 2.1), we sought to do *exactly* this. That is, we sought to guide the field toward several of the broad existing practices, many of which you mention, e.g.,
> * An emphasis on adaptive attacks (criterion D1)
> * Nominal performance and efficiency trade-offs (criteria D2-D3)
> * Black-box vs. white-box comparisons (criterion D4)
>
> As far as we know, **this is the only paper that attempts to provide such a "guide" in the field of jailbreaking**, and we hope that you agree that this constitutes a valuable contribution. We look forward to discussing this with you.
>
> ---
>
> ## Theoretical guarantees
>
> > "I don't know why k-unstability would be a fundamental property of adversarial suffix attacks like GCG. It's an empirical observation for a particular attack but we know very well now that building defenses on attack-specific artifacts is not a way to go."
>
> We respectfully disagree with several points here.
>
> * **"Fundamental."** We never claim that $k$-instability is a "fundamental property" of GCG. It is a simplifying assumption, based on which we were able to prove Prop. 4.1.
> * **"Empirical observation."** This observation also significantly improves robustness against semantic attacks. See Appendix H in our updated PDF, wherein we show that SmoothLLM significantly improves robustness against the PAIR semantic jailbreak [A], setting a new state-of-the-art.
> * **"Know very well."**  The claim in the second sentence relies on the assumption that jailbreaking will follow the same trends as the AE literature. We are not aware of any evidence that supports this claim.
>
> If you would like us to refer to $k$-instability as a "simplifying" assumption rather than "mild" (as we do in the second paragraph of Section 4) throughout our paper, we would be happy to add this language.
>
> > "It only leads to a counterproductive arms race without yielding a long-lasting idea."
>
> This claim is indeterminate. We hope you agree that there is no way that we, nor anyone else, can accurately predict what will constitute a "long-lasting idea," especially given that the jailbreaking field is very young.
>
> > "Considering that it would be generally easy for an attacker to detect if SmoothLLM defense is used (e.g., by looking at the model's responses), it also would be trivial to change up the attack strategy to break the defense."
>
> We do not agree with these claims.
> * **"Detect."** As far as we know, there is no clear way for an adversary to detect whether SmoothLLM is being used, and moreover a paper has never been written on this topic.
> * **"Trivial."** We are not aware of any way to "change up" the attack strategy to break the defense. No paper has been written on this topic.
>
> Both claims are unsupported by *any* evidence. If you can provide evidence to support these claims, we would be happy to discuss further.
>
> > "Is there a reason why you believe k-unstability is a good assumption that will withstand future attacks or stronger adaptive attacks?"
>
> In our paper, we considered an existing and well-established threat model, and developed a theory that fits within this threat model. This theory contributed to the development of SmoothLLM, which (1) withstands GCG attacks (see Section 5), (2) withstands adaptive GCG attacks (see Appendix C), and (3) significantly reduces the success rate of the PAIR semantic jailbreaking attack (see Appendix H in our updated PDF). As we cannot anticipate what future attacks will be, we hope you agree that there is no way we can anticipate how SmoothLLM will fare against them.

---

> ### Author Response · Authors · 2023-11-16
> **Response Part 2: Adaptive attacks and our choice of threat model**
>
> ## Adaptive attacks
>
> > "It's accepted now that adaptive attack evaluations should be at the forefront of defensive papers like this. . . I can't see much deeper thought in this paper in this regard."
>
> This is incorrect.  There is no standard practice regarding adaptive attacks in the jailbreaking literature, especially given that SmoothLLM is the first defense paper in this field.  In this vein, we sought to guide the community toward considering adaptive attacks (see criteria D1 in Section 2.1).  This is why we devoted three pages in Appendix C to deriving an adaptive attack for the GCG threat model.  This took substantial effort, thought, and weeks of experimentation, and as far as we know, **this is the only adaptive jailbreaking attack in the literature.**  And crucially, Figure 12 demonstrates that **SmoothLLM is highly robust to the adaptive attack we derived**.
>
> > "It definitely needs more work to be convincing that this defensive strategy is not going to be broken by the next deadline."
>
> We do not agree.  In Figure 12, we provide concrete evidence that SmoothLLM is not susceptible to adaptive attacks.  Furthermore, we have no way of predicting whether this defense will be broken by the next deadline.  Could you be more specific regarding why you find this to be unconvincing?
>
> > It's trivial to defend against any particular attack (e.g., by detecting the attack's artifacts) but through many painful lessons, we know that adaptive attacks are hard to design and evaluate against.
>
> Again, we respectfully disagree.  **Until our paper, there was no evidence that the GCG attack could be mitigated.**  Even now, four months after GCG was proposed, companies like OpenAI, Google, and Anthropic have not yet been able to mitigate this attack (see the next section of this rebuttal), which is strong evidence supporting the fact that defending against GCG is not "trivial."  Therefore, the fact that SmoothLLM can reduce the attack success rates w/r/t both adaptive and non-adaptive versions of GCG is a core contribution, which we hope will help the field avoid some of the "painful lessons" you mention.
>
> ---
>
> ## Threat models
>
> > "With attacks like GCG, I think, we might be making the same mistake and focusing on threats that might not matter."
>
> We respectfully disagree: There is no evidence that GCG "might not matter." On the contrary, we argue that it matters just as much as any other jailbreak.  Consider the following timeline.
>
> * **July 27, 2023: GCG appears on arXiv.** Table 1 in the GCG paper showed that GCG was frequently able to jailbreak ChatGPT, Claude, and PaLM.
> * **Sept. 28, 2023: ICLR submission deadline.** Figure 1 in our paper shows that as of the submission deadline, OpenAI, Anthropic, and Google were unable to mitigate GCG attacks.
> * **Nov. 14, 2023: ICLR rebuttal period.** We reran the experiment described by Figure 1 in our paper and we got nearly identical results.
>
> This evidence indicates that GCG *still* consistency jailbreaks state-of-the-art LLMs, including ChatGPT, which has nearly 200 million active users.  Resolving this highly realistic threat is essential and evidently, as the jailbreak still works as of this week, nontrivial.  Given this, could you explain why you feel that the GCG threat model is "trivial" and "non-realistic?"
>
> > "There’s some real-world evidence that semantic jailbreak prompts are a greater and harder-to-mitigate risk that attackers are already deploying [1]"
>
> Again, we respectfully disagree with several of the points raised here.
>
> * **"Greater risk".** Token-level jailbreaks like GCG and semantic jailbreaks like DAN [1] both pose the same risk: Both can jailbreak state-of-the-art LLMs.
> * **"Harder-to-mitigate risk."**  We are not aware of any published literature that conclusively proves that semantic jailbreaks are "harder-to-mitigate."  **The only way to determine which kinds of jailbreaks are harder to defend against is to publish papers like SmoothLLM**, which demonstrates how one can mitigate specific jailbreaking algorithms.  This is precisely the contribution of our paper.
> * **"Already deployed."** GCG and related token-based variants *are* "already deployed." They have been used to jailbreak LLMs, elicit hallucinations [B], produce toxic text [C].
>
> > There's an intellectual merit in studying more academic, theoretical threats but I think it is critical to guide the community towards studying real threats. I can't see this paper doing that.
>
> We do not understand this sentence.  You seem to draw a line between "theoretical" and "real" threats, but it's unclear to us why GCG is "academic" and DAN is "real."  Both algorithms pose very real threats, in the sense that they can jailbreak state-of-the-art LLMs.  Could you be more specific about how you draw this line, and point to references that back up this sentiment?

---

> ### Author Response · Authors · 2023-11-16
> **Response Part 3: Answering your additional questions**
>
> > "There’s no discussion on whether this defense would work against semantic jailbreaks or other real-world strategies, and no discussion on the limitations of the defense in deployment.""
>
> As mentioned above, GCG *is* a "real-world strategy," in the sense that it extracts *real* harmful content out of commonly used LLMs like ChatGPT, which has hundreds of millions of active users.  However, we'd be happy to add a discussion regarding the applicability of SmoothLLM to semantic jailbreaks.  In fact, we ran SmoothLLM on prompts generated for Vicuna using PAIR, a recently proposed semantic jailbreak [A].  We found that **SmoothLLM reduces the ASR of this jailbreak by as much as 40%** (see Appendix H in our updated PDF).  As far as we know, this sets the state-of-the-art in defending against the PAIR jailbreak. We hope that you agree that this broadens the contribution of our paper.
>
> ---
>
> ## Additional questions
>
> > "I'm not entirely sure about the reasoning behind returning one of the responses R_j (Line 5-6 in the Alg 1). How did you make this decision? What are the downsides compared to the other options you discussed in Section 6? I'm failing to have good intuition about this. Could you clarify?"
>
> We view SmoothLLM as a wrapper around any vanilla black-box LLM (see Figure 2).  In this way, given a prompt as input, we designed SmoothLLM to return a *single* response as output.  Since smoothing generally relies on *voting* to determine the output, we found it natural to return a response consistent with the majority vote.
>
> We are excited about improving SmoothLLM and extending it to different problem settings in future work.  Here is what we had in mind for each of the other variants listed at the end of Section 6 (in the order in which we mentioned them).
> * **Abstaining when an attack is detected.** This would be useful if you want to *ensure* that harmful content never reaches the user.  However, the downside is that refusing to generate any text when a jailbreak is detected would complicate the process of evaluating the ASR and result in unnecessary conservatism, which is why we used the voting approach in the paper.
> * **Generating based on unperturbed prompts.** This idea could be used to improve the nominal/clean performance.  As jailbreaks are unlikely to be detected for "clean" prompts, this may result in better trade-offs.  However, this also makes the LLM more vulnerable to attacks, since malicious prompts that evade the detector could be processed in their unperturbed forms, potentially exposing the user to objectionable content.
> * **Denoising perturbed prompts.** Denoising the perturbed prompts could change the adversarial suffixes enough to render the attack benign, and this could also improve clean/nominal performance.  However, there is evidence that such techniques are highly susceptible to adaptive attacks (see, e.g., [D]), which is why we didn't go down this road.
>
> We hope this gives you insight into our design process for SmoothLLM.  And all this being said, we are excited about studying each of these variants in future work.
>
> > "How does the defense affect the model's response utility against very short prompts (which might have less redundancy to withstand perturbations)?"
>
> On average, goal prompts in the harmful behaviors dataset are 72 characters long, and suffixes are ~100 characters long (see Table 2).  As goal prompts get shorter, perturbations to the prompt are more likely to degrade the semantic content since they contain less redundant information (as you mention), which could potentially weaken the robustness vs. nominal performance trade-off.  However, as long as the GCG attack stays fixed, shorter goal prompts mean that perturbations are more likely to affect the suffix, which would mean that smaller values of $q$ could be used, which would improve the aforementioned trade-off.  Furthermore, it's difficult to write prompts that request toxic content in fewer than ~50 characters anyway, so this case isn't particularly concerning.
>
> > "Do you use the input-specific GCG in your experiments?"
>
> Yes, we use the instance-specific version.
>
> > "What about the universal version of GCG?"
>
> On the one hand, this version requires significantly more compute, as it essentially requires running an algorithm with similar complexity to the instance-specific version for $M$ goal prompts.  On the other hand, as you mention, it only needs to be run once.  We expect that this version is just as unstable as the instance-specific suffixes.  We are running this experiment now, and will let you know when it completes and what conclusions we can draw.
>
> ---
>
> [A] "Jailbreaking black box large language models in twenty queries"
> [B] "LLM Lies: Hallucinations are not Bugs, but Features as Adversarial Examples"
> [C] "Automatically Auditing Large Language Models via Discrete Optimization"
> [D] "Language Model Detectors Are Easily Optimized Against"

---

### Official Review · Reviewer_rsea · 2023-11-02

**Soundness:** 2 fair
**Presentation:** 3 good
**Contribution:** 2 fair
**Rating:** 3
**Confidence:** 5

**Summary:**

In this paper, the authors proposed SmoothLLM to mitigate jailbreaking attacks on LLMs. The method is based on the finding that adversarially-generated prompts are brittle to character-level changes. Specifically, the defense first randomly perturbs multiple copies of a given input prompt, and then aggregates the corresponding predictions to detect adversarial inputs.

**Strengths:**

1. found that adversarially-generated prompts are brittle to character-level changes
2. proposed a new algorithm for defending against jailbreaking attacks in llm
3. The main idea follows randomized smoothing in image domain and provide some theoretical results

**Weaknesses:**

1. The “robustness guarantee” that generalizes the original randomized smoothing to the LLM setting in this paper does not seem to be a valid “guarantee”, as it actually depends on some unverifiable assumption (k-unstable). Therefore, different from those traditional robustness guarantees, where one could verify that some examples must be robust, the “guarantee” in this paper cannot provide any real certified robust accuracy. In this sense, I don’t think the provided theorem provides any type of formal guarantee to robustness here. It seems a bit misleading to call it a formal robustness guarantee.

2. From the algorithm design, the proposed algorithm can easily have degraded nominal performances (since the output is randomly sampled from the perturbed input’s response and each perturbed input has changed the input quite a lot). Although the authors also consider that as one of the major aspects to test, I don’t see that part and the corresponding experiments very convincing.  Table 3 actually suggests that when N is a bit large, the nominal performance significantly drops. More importantly, it only tested on light perturbation cases where q <= 5%, while in the main experiments, most were conducted in the case where q = 10/15/20%. This makes me very concerned about the actual use case of the proposed algorithm.

3. The discussion on the efficiency part seems also misleading. All the comparison listed in this paper seems to compare the SmoothLLM with GCG, which is totally unreasonable to me (take an analogy, this is similar to comparing the complexity of randomized smoothing and adversarial attack). I didn’t see this comparison has too much meaning here. Since SmoothLLM is basically an inference-time defense, shouldn’t the authors compare it with normal LLM inference? And that seems to be a solid N times larger than the normal inference in my opinion. Since the inference time is one major obstacle in modern LLM deployment, SmoothLLM does not really seem efficient at all.

4. One important aspect that seems to be missing is the adversarial prompt length. Essentially in this setting, there is no traditional invisible adversarial length constraint (like in traditional adversarial examples, usually we are only allowed to change one/few works/characters). Basically, your adversarial prompt can be any length, as long as it successfully breaks the alignment. However, I am a bit concerned that under such situations where the proposed attack still work? From the theoretical guarantee, I didn’t thoroughly check the proof but the original randomized smoothing cannot provide meaningful results when the adversarial part is large. From the empirical perspective, if we just programmed GCG to generate a long adversarial prompt (not the only way but you can imagine many different ways to achieve the goal) would it still work? Or it still work, but may need a significantly larger perturbation ratio q and thus lead to degraded nominal performances?

**Questions:**

1. I do not quite understand why the authors claimed that for the efficiency objective, “SmoothLLM is independent of the prompt length”?

---

> ### Author Response · Authors · 2023-11-16
> **Response Part 1: What is meant by "robustness guarantees," clarifications regarding non-conservatism**
>
> ## Robustness guarantees
>
> > "The 'robustness guarantee' that generalizes the original randomized smoothing to the LLM setting in this paper does not seem to be a valid 'guarantee', as it actually depends on some unverifiable assumption (k-unstable)."
>
> This is incorrect in several senses.
> * **"Generalize."** Prop. 4.2 does not "generalize" randomized smoothing (RS).  RS and SmoothLLM are different algorithms designed for different problems; neither generalizes the other.
> * **"Unverifiable."** The $k$-instability assumption is not "unverifiable."  It is verifiable, although checking whether it holds would take exponential time in the worst case.
> * **"Valid guarantee."** Unless you can point to a mistake in our proof, our guarantee in Prop. 4.2 is completely sound and valid. That you find the assumption of $k$-stability to be difficult to check does not impact whether the result is valid.
>
> If you disagree with any of these bullets, we would be glad to discuss further.
>
> > ". . .different from those traditional robustness guarantees, where one could verify that some examples must be robust, the “guarantee” in this paper cannot provide any real certified robust accuracy."
>
> You are making a false comparison between two distinct fields: adversarial examples (AEs) and jailbreaking.  As discussed in Appendix E.2, these fields involve distinct problem settings and threat models, which necessitate different kinds of robustness guarantees. Although RS and SmoothLLM both use "smoothing" techniques, our goal is *not* to replicate "traditional robustness guarantees" from the AEs literature.  Rather, we prove a guarantee in Prop. 4.2 that makes sense in the context of jailbreaking.  Furthermore, to be clear: **We never claim to provide a robustness *certificate***; the words "certified" and "certificate" do not appear in our paper.
>
> > "In this sense, I don’t think the provided theorem provides any type of formal guarantee to robustness here. It seems a bit misleading to call it a formal robustness guarantee."
>
> We do not agree.  Our result is formal and provides a guarantee on robustness under the assumption of $k$-instability; we describe Prop. 4.2 in precise langauge and formally prove that it holds in Appendix A.  Could you explain why you think our result is not a "formal robustness guarantee?"
>
> ---
>
> ## Non-conservatism
>
> > "From the algorithm design, the proposed algorithm can easily have degraded nominal performances. . . I don’t see that part and the corresponding experiments very convincing."
>
> Regarding your being unconvinced, here is what the evidence says:
>
> * For Vicuna, if you set $q=5\%$, set $N=20$, and use swap perturbations, sacrificing between $5$-$10\%$ in nominal performance (see Table 4) reduces your susceptibility to jailbreaking from $98\%$ to around $10\%$ (see Figure 7).
>
> Therefore, as is the case in nearly every paper ever written on robustness in ML, improving robustness comes at the (relatively small) cost of trading off some nominal performance.  **Our paper is the first to study this trade-off for jailbreaking, and we feel that this constitutes a notable contribution**.  If you still disagree, could you be more specific about what you find to be unconvincing?
>
> > "Table 3 actually suggests that when N is a bit large, the nominal performance significantly drops."
>
> This is incorrect.  In every nearly every experiment in Table 4, nominal performance improved as $N$ was increased from 2 to 20.
>
> > "This makes me very concerned about the actual use case of the proposed algorithm."
>
> The goal of our paper is not to solve jailbreaking in one shot by designing an algorithm that is without any drawbacks.  Rather, our goal is to contribute to a field of study that will, we believe, lead to the discovery of strong defenses like SmoothLLM. And in particular, while there are trade-offs, the "actual use case" for SmoothLLM is this:
>
> * **SmoothLLM is the first algorithm that provides *any* robustness against GCG attacks.**
>
> While there may be trade-offs and/or aspects of the algorithm that could be improved in future work, we strongly feel that our paper advances research on this topic.  Do you disagree?

---

> ### Author Response · Authors · 2023-11-16
> **Response Part 2: Clarifications regarding efficiency and the adversarial suffix length**
>
> ## Efficiency
>
> > "Since SmoothLLM is basically an inference-time defense, shouldn’t the authors compare it with normal LLM inference?"
>
> This comparison is provided in our paper.  Undefended LLMs are $N$ times more query efficient than SmoothLLM.  See the paragraph in Section 3 titled "Aggregation step."
>
> > "The discussion on the efficiency part seems also misleading. All the comparison listed in this paper seems to compare the SmoothLLM with GCG, which is totally unreasonable to me."
>
> In Section 3, we compared the efficiency of undefended LLMs and SmoothLLM.  Following this, we thought that it would be valuable to provide another perspective on efficiency, so in Section 5, we compared the query efficiency of the attack (GCG) to our defense (SmoothLLM). We find it interesting that although creating a GCG suffix requires 256,000 queries, defending the corresonding attack takes only ~10 queries.
>
> If you think it would make our paper stronger, we would be happy to move this ablation study to the appendix, but we feel strongly that it is not a weakness of our paper.  If you disagree, could you explain in more detail why you think this comparison is "totally unreasonable" and "misleading" given that we already provided the comparison you asked for?
>
> > "Since the inference time is one major obstacle in modern LLM deployment, SmoothLLM does not really seem efficient at all."
>
> You are correct that SmoothLLM is less query efficient than normal LLM inference.  As we all know, when it comes to robustness, there is usually no free lunch, and indeed our work shows that there is a trade-off between efficiency and robustness.  We hope you agree that it's unreasonable to expect a defended algorithm to have exactly the same efficiency as an undefended LLM.
>
> Furthermore, here's a point of comparison.  For $\ell_2$ randomized smoothing in the setting of adversarial examples, one of the seminal papers (Cohen et al., 2019) uses $N=100,000$ random samples.  In the separate setting of jailbreaking, we show that SmoothLLM obtains strong robust for $N=10$ samples.  Therefore, one contribution of our work is to show that the robustness-efficiency trade-off is far milder in jailbreaking than it is for adversarial examples.
>
> ---
>
> ## Adversarial suffix length
>
> > "From the theoretical guarantee, I didn’t thoroughly check the proof but the original randomized smoothing cannot provide meaningful results when the adversarial part is large."
>
> This comment highlights a strength of our method.  Unlike the adversarial examples version of randomized smoothing, increasing the adversarial budget (i.e., the suffix length) will actually make the defense stronger, since the random perturbations will have a better chance of hitting the suffix.  For details, see Lemma A.2.
>
> > "From the empirical perspective, if we just programmed GCG to generate a long adversarial prompt. . . would it still work?"
>
> Yes, since $q$ is a percentage, SmoothLLM is agnostic of the prompt length.  In fact, as discussed above, as the suffix length increases, the robustness of SmoothLLM will improve.
>
> > "One important aspect that seems to be missing is the adversarial prompt length."
>
> We would be happy to add the above discussion to the paper if you feel it would make our paper stronger.

---

### Official Review · Reviewer_Utne · 2023-11-05

**Soundness:** 3 good
**Presentation:** 3 good
**Contribution:** 2 fair
**Rating:** 5
**Confidence:** 4

**Summary:**

This paper proposes a defense algorithm to mitigate jailbreaking attacks on LLMs. It works by first randomly perturbing the input prompt (via insert, swap or batch), and then conducting a majority voting of the resulting predictions to detect adversarial inputs. Provable guarantees are also provided on attack mitigation. Experiments show that the defense reduces ASR and maintains a certain utility on clean inputs.

**Strengths:**

1. Defending against jailbreaking attacks of LLM is an important problem for trustworthy LLMs in practice;

2. The proposed method adapts the randomized smoothing principle to LLM, and conducted extensive evaluation to empirically demonstrates its ability for defending jailbreaking attacks;

3. The paper presentation is clear and easy-to-follow.

**Weaknesses:**

1. The major concern is that perturbing the prompts could greatly influence the LLM’s original behavior. The provided evaluation of non-conservatism is only based on rather simple tasks (i.e., classification), which does not verify whether the LLM can still have normal generation behavior on randomly perturbed prompts.

2. The proposed method is based on the observation that adversarial suffixes are fragile to character-level perturbations, ignoring the (un)stability of normal prompts to such perturbation. The paper did not draw a clear boundary (either theoretically or empirically) of how perturbation only destroys the adversarial suffix and maintains the semantics of normal ones.

3. The theoretical guarantee relies on the assumption, k-stable, that cannot be verified or calculated in practice (unlike the assumption in randomized smoothing), and can hardly be “realistic”. The resulting theoretical guarantee thus is not really a rigorous one. For instance, Figure 6 is a conceptual result, instead of a guarantee calculated on real prompts.

4. While the paper has compared its method with randomized smoothing (e.g., image v.s. text), it is still a straightforward application of the perturbation-then-voting principle, which limits its technical novelty.

5. The paper has several statements that could be over-claimed and misleading, regarding “the first defense”, theoretical results and empirically conclusions. See details in Questions.

**Questions:**

1. As said in weakness 1, character-level perturbation may destroy prompt semantics and confuse LLM. Therefore I am concerned with the true cause of the decreased ASR by the proposed method. It may not necessarily be the result of destroying the adversarial suffix, instead, it is possible that the perturbation destroys the semantics of the malicious question, thus the LLM model responds with something like “Sorry, I cannot understand”. Did the authors observe such cases? And a possible evaluation to verify such a case: using the behavior dataset without adversarial suffix (i.e., only keeping the malicious question part), then using an unaligned version of LLM, and checking its ASR drop after random perturbation.

2. The provided non-conservatism evaluation is insufficient to really validate that SmoothLLM maintains nominal performance, because: 1) these tasks are simple (e.g., classification tasks); 2) the accuracy drop is actually relatively large even when q is set small and N is large (e.g., random guess on PIAQ is already 50, but SmoothLlama2 with q=2 and N=6 gives 59, which is a large drop compared with original performance 76.7). Can the authors provide a more convincing evaluation of nominal behavior of LLMs, e.g., on generation tasks?

3. In table 3 of the non-conservatism evaluation, are the numbers the average results of all N samples? What are the variances?

4. As said in weakness 2, SmoothLLM is built based on the observation in Figure 4 that adversarial suffixes are fragile to character-level perturbations. Then an implicit assumption is that clean prompts are more stable to such perturbations. Can the authors provide corresponding evidence to verify (in)stability of clean prompts, similar to Fig. 4?

5. The theoretical proof is provided for swap and batch perturbations, instead of “other perturbation types”. Is the guarantee/proof applied to the insert perturbation?

6. The paper claims the efficiency of SmoothLLM by comparing the runtime with attacks (e.g., GCG). This seems to be an unreasonable comparison as one is defense and other is attack. Can the authors soften this claim and focus on the comparison of SmoothLLM and vanilla LLM (and other defenses)?

7. The paper emphasizes in multiple places that this is the first algorithm to defend LLM jailbreaking attacks, which could be over-claimed as several attempts have been proposed [1, 2]. Can the authors discuss them and adjust corresponding claims if this concern makes sense?

[1] Kumar, Aounon, et al. "Certifying llm safety against adversarial prompting." arXiv preprint arXiv:2309.02705 (2023).

[2] Jain, Neel, et al. "Baseline defenses for adversarial attacks against aligned language models." arXiv preprint arXiv:2309.00614 (2023).

---

> ### Author Response · Authors · 2023-11-16
> **Response Part 1: Clarifications regarding non-conservatism and our theoretical guarantees**
>
> ## Non-conservatism
>
> > The provided evaluation of non-conservatism is only based on rather simple tasks (i.e., classification), which does not verify whether the LLM can still have normal generation behavior on randomly perturbed prompts. . . Can the authors provide a more convincing evaluation of nominal behavior of LLMs, e.g., on generation tasks?
>
> We disagree.  We used three well-established benchmarks---TruthfulQA, ToxiGen, and PIQA---which are commonly used to evaluate the performance of LLMs.  **All three datasets were used in the Llama2 paper [A].**  Beyond doing a human evaluation, we are not aware of any datasets that satisfactorily measure "normal generation," or what it would mean for any generated text to be "normal."  If you have a dataset in mind that measures generation, we would be happy to evaluate our method on it.
>
> > ". . .character-level perturbation may destroy prompt semantics and confuse LLM. Therefore I am concerned with the true cause of the decreased ASR by the proposed method. It may not necessarily be the result of destroying the adversarial suffix, instead, it is possible that the perturbation destroys the semantics of the malicious question, thus the LLM model responds with something like “Sorry, I cannot understand”. Did the authors observe such cases?"
>
> Your question was addressed in our original submission.  We encourage the reviewer to look at Appendix D---entitled "The Incoherency Threshold"---which shows an example wherein the LLM cannot understand the perturbed prompt, and thus responds with "I must inform you that the statement you provided is not factually coherent and does not make sense to me.  The question contains a series of unrelated words and symbols that do not form a valid question."  We provide guidelines for how to avoid this scenario in Appendix D.  If the reviewer has any questions about this, we would be happy to discuss further.
>
> > And a possible evaluation to verify such a case: using the behavior dataset without adversarial suffix (i.e., only keeping the malicious question part), then using an unaligned version of LLM, and checking its ASR drop after random perturbation.
>
> This is an interesting suggestion!  Do you have any suggestions for unaligned LLMs?  We would be happy to run this experiment if you feel that it will make our paper stronger.
>
> > "In table 3 of the non-conservatism evaluation, are the numbers the average results of all N samples? What are the variances?"
>
> We think that there might be a misunderstanding.  As you rightly said elsewhere in your review, our method uses a "perturbation-then-voting principle."  The figures in Table 3 use the same SmoothLLM algorithm (i.e., voting), and so there are no variances to report.
>
> > "The proposed method is based on the observation that adversarial suffixes are fragile to character-level perturbations, ignoring the (un)stability of normal prompts to such perturbation. . . Can the authors provide corresponding evidence to verify (in)stability of clean prompts, similar to Fig. 4?"
>
> We are confused by this question.  This evidence is provided in Table 3 of our paper.  Could you be more specific about the evidence that you would like us to provide?
>
> ---
>
> ## Theoretical guarantees
>
> > The theoretical guarantee relies on the assumption, k-stable, that cannot be verified or calculated in practice (unlike the assumption in randomized smoothing), and can hardly be “realistic”.
>
> This is incorrect.  The $k$-instability assumption can be calculated on real prompts, although checking whether it holds would take exponential time in the worst case.  However, the difficulty involved in checking this assumption doesn't impact whether it is realistic; Figure 4 provides empirical evidence that this assumption is realistic for real prompts.  If you would like us to change the word "realistic" to "simplifying," we would be happy to do so.
>
> >The resulting theoretical guarantee thus is not really a rigorous one. For instance, Figure 6 is a conceptual result, instead of a guarantee calculated on real prompts.
>
> We disagree with this for several reasons.
> * **"Rigorous" is not the opposite of "conceptual."**  Unless you can point to a mistake in our proof, we hope you agree with us that Prop. 4.1 is completely sound and rigorous.
> * **"Real prompts."** Our result *can* be calculated on "real prompts," although in the worst case, computing $k$-instability may take exponential time.
>
> If you disagree with either of these points, we would be happy to discuss further.
>
> > The theoretical proof is provided for swap and batch perturbations, instead of “other perturbation types”. Is the guarantee/proof applied to the insert perturbation?
>
> By "other perturbation types," we meant *patch* perturbations, since we did not have space to fit the patch perturbation proposition statement in the main text.  Do you feel that including the insert perturbations proof would make our paper stronger?  If so, we would be happy to provide it.

---

> ### Author Response · Authors · 2023-11-16
> **Response Part 2: Clarifications regarding efficiency, novelty, and concurrent work**
>
> ## Efficiency
>
> > The paper claims the efficiency of SmoothLLM by comparing the runtime with attacks (e.g., GCG). This seems to be an unreasonable comparison as one is defense and other is attack. Can the authors soften this claim and focus on the comparison of SmoothLLM and vanilla LLM (and other defenses)?
>
> We disagree with several points here.
>
> * **"Vanilla LLMs."**  A comparison against vanilla LLMs is already provided in our paper.  Vanilla LLMs are $N$ times more query efficient than SmoothLLM.  See the paragraph in Section 3 titled "Aggregation step."
> * **"Unreasonable comparison."** After comparing to vanilla LLMs, we thought that it would be valuable to provide another perspective on efficiency, so in Section 5, we compared the query efficiency of the attack (GCG) to our defense (SmoothLLM). We find it interesting that although creating a GCG suffix requires 256,000 queries, defending the corresponding attack takes only ~10 queries.  If you like, we could move this ablation study to the appendix.
> * **"And other defenses."** As we discuss in the "Concurrent work" section of this rebuttal, there are currently no published defenses on this topic.  [1] and [2] first appeared on arXiv three weeks before the submission deadline, and therefore it is unreasonable to ask us to compare to them.
>
> We would be happy to discuss any/all of these points further.
>
> ---
>
> ## Novelty
>
> > "While the paper has compared its method with randomized smoothing (e.g., image v.s. text), it is still a straightforward application of the perturbation-then-voting principle, which limits its technical novelty."
>
> The novelty of our approach is in adapting classical ideas from the smoothing literature to the new and pressing problem of jailbreaking LLMs.  While smoothing is an existing technique, we hope you agree that **our paper is the first to show that the attack success rate of GCG can be significantly decreased** via a smoothing-based technique, which is (in our opinion) a highly novel contribution.
>
> ---
>
> ## Concurrent work
>
> > "The paper emphasizes in multiple places that this is the first algorithm to defend LLM jailbreaking attacks, which could be over-claimed as several attempts have been proposed [1, 2]. Can the authors discuss them and adjust corresponding claims if this concern makes sense?"
>
> We appreciate you pointing us toward these papers.  However, please consider the following:
> * [2] appeared on arXiv on Sept. 1, 2023.
> * [1] appeared on arXiv on Sept. 6, 2023.
> * The ICLR submission deadline was Sept. 28, 2023.
> * Both [1] and [2] are currently under review at ICLR.
>
> We hope the reviewer agrees with us that (a) **it is unreasonable to ask us to compare to works that appeared on arXiv *three weeks before* as the ICLR submission deadline** and (b) **these works should be viewed as concurrent to our paper**.  If you would like us to specify that these works are concurrent in our main claims, we would be happy to do so.
>
> Moreover, here is a summary of these two works, which we would be happy to add to our related works section:
>
> * In [1], the authors consider several candidate defenses, including preprocessing using a perplexity filter, paraphrasing input prompts, and employing adversarial training.  Results for these methods are mixed; while heuristic detection-based methods perform strongly, adversarial training is shown to be infeasible given the computational cost of retraining LLMs.
> * In [2], the authors propose a technique which provides certifiable robustness guarantees by applying a safety filter on sub-strings of input prompts.  While promising, the complexity of this method scales with the length of the input prompt, which is undesirable.  Moreover, unlike our work, [2] does not evaluate their proposed defense against GCG attacks.
>
> ---
>
> [A] Touvron, Hugo, et al. "Llama 2: Open foundation and fine-tuned chat models." arXiv preprint arXiv:2307.09288 (2023).

---

### Author Response · Authors · 2023-11-21
**A summary of our rebuttal; we would appreciate your feedback before the end of the rebuttal period**

Hello reviewers! Thanks again for your thoughtful comments on our paper. We recognize that the reviewing window is short this year, but given that (a) we posted our rebuttals five days ago and (b) we feel confident that we have addressed most (if not all) of your concerns, we would greatly appreciate your feedback once more.

**The strengths of SmoothLLM.** Here are the reasons you have given to accept our paper:
- **Critical problem setting.** Our paper addresses a critical problem: ChatGPT has 200 million active users, and the fact that this LLM (and others) can be jailbroken by GCG is of enormous concern. You called our setting "an important problem" (`Utne`) and an "early exploration" (`qFYD`).
- **Novelty.** Our paper is the first (modulo concurrent works) to propose a defense (namely, SmoothLLM) against jailbreaking attacks. You all agreed with this; in your own words, you called our work "important" (`Utne`), "new" (`rsea`), "critical" (`Ay91`), and "pioneering" (`qFYD`).
- **Extensive experiments.** We show empirically that SmoothLLM empirically reduces the attack success rate of GCG to below 1%. You highlighted our "extensive experiments" (`Utne`) and said that "[our] guidelines and experiments. . . are insightful" (`Ay91`).
- **Theoretical insights.** We are the first to provide *any* theoretical results for this problem setting. You agreed, saying that our theory informed when SmoothLLM is "effective" (`Ay91`).

**Your concerns.** Here are the main concerns that you highlighted in your reviews:
1. **Theoretical assumption.** Our theoretical result (Prop. 4.1) has an assumption that is expensive to verify (`Utne`, `rsea`, `Ay91`).
2. **Trade-offs.** Our algorithm incurs a trade-off between robustness, query efficiency, and nominal performance (`Utne`, `rsea`).
3. **Concurrent works.** There are other concurrent works that defend against jailbreaking attacks (`Utne`).
4. **Adaptive attacks.** Our paper focuses on both adaptive and non-adaptive attacks (`Ay91`).
5. **Threat model.** The threat model for GCG relies on knowing the target string (`qFYD`).

We believe that none of these concerns are critical, as we explain below.

**Our responses.** Here is short summary of our responses to your concerns:
1. **Theoretical assumption.** Definition 4.1 is a *simplifying* assumption, and does not impact the soundness of our results. While it may be expensive (although, certainly possible) to check in practice, Fig. 4 gives qualitative evidence for this assumption holding for GCG attacks.
2. **Trade-offs.** As is the case in nearly every paper ever written on robust ML, improving robustness comes at a *relatively small*, though nonzero cost in nominal performance and query efficiency. Our transparency regarding these trade-offs, in conjunction with the fact that we are the first to study these trade-offs in the context of jailbreaking, should be seen as a contribution, not a weakness.
3. **Concurrent works.** We added a discussion of the two concurrent papers to our related work; all three of these works should be seen as the *first* to defend against jailbreaking attacks. As both papers appeared on arXiv less than a month before the ICLR deadline, and as neither has a public code repository, we are unable to compare to either paper.
4. **Adaptive attacks.** We are the first to derive an adaptive jailbreaking attack for LLMs. This is not a limitation; it is a contribution of our paper, and the fact that SmoothLLM is resistant against such an attack is another significant strength of SmoothLLM.
5. **Threat model.** We clarified that the threat model we considered does not depend on the target string. This was a minor bug in our presentation, and now that it is fixed, it should not impact our acceptance/rejection decision.

**What we added.** Here's a summary of what we added to our paper during the rebuttal period; see the *green* text in our updated PDF.
* **Defending against semantic jailbreaks.** In Appendix H (Figure 17), we showed that SmoothLLM reduces the ASR of the PAIR *semantic* jailbreak from 92% to ~50%. This is the first and only demonstration of the defending against semantic jailbreaks in the literature, and therefore we believe that this constitutes a significant contribution.
* **Discussion of concurrent works.** In Section 2, we added a discussion of two concurrent works which deal with defending against jailbreaking attacks. Both papers appeared on arXiv *within one month* of the ICLR deadline, and both are under review at this conference.
* **Clarification of the definition of jailbreaks.** In Section 2, we clarified the definition of what constitutes a jailbreak. This *completely* resolves all of the concerns raised by `qFYD`.
* **Correction of typos.** We corrected a couple of minor typos. Thanks to `qFYD` for pointing these out.

Please let us know what you think about (a) our rebuttal and (b) the changes we made to our paper in line with your feedback. Thanks again!

---

> ### Author Response · Authors · 2023-11-22
> **Today (Wed., Nov. 22) is the last day of the discussion period**
>
> Hi reviewers!
>
> Today is the last day of the discussion period. If you have a chance, please read through our rebuttal and let us know whether we have addressed your concerns. We greatly appreciate your ongoing engagement with our paper and the feedback you have provided thus far.
>
> Thanks,
> The authors

---

### Meta-Review · Area_Chair_m2Cv · 2023-12-06

**Metareview:**

The paper presents a novel defense mechanism, SmoothLLM, against jailbreaking attacks on LLMs. Despite its innovative approach, the paper has significant gaps in both theoretical underpinnings and practical evaluations. Key concerns include the reliance on the questionable k-unstability assumption, limited and unconvincing empirical evaluations, and a lack of consideration for adaptive attacks and real-world applicability. Also there has been overstatements regarding the novelty and effectiveness of the proposed method.

**Justification For Why Not Higher Score:**

The paper's main idea of k-unstability isn't really backed up well enough; it's a shaky assumption that doesn't hold up in all cases. The tests they did to show how well the LLM works with their method weren't that convincing, especially for complex real-life tasks. Comparing their defense method's speed with attack methods doesn't make much sense Plus, they didn’t dig into how well this would work against smarter, changing attacks, which is important for real-world use.

**Justification For Why Not Lower Score:**

NA

---

### Decision · Program_Chairs · 2024-01-16

Reject